# Topics of Study in Under-18 Padel Categories: A Scoping Review

**DOI:** 10.3390/sports13030075

**Published:** 2025-03-04

**Authors:** Iván Martín-Miguel, Diego Muñoz, Rafael Conde-Ripoll, Álvaro Bustamante-Sánchez, Bernardino J. Sánchez-Alcaraz, Adrián Escudero-Tena

**Affiliations:** 1Faculty of Sport Sciences, University of Extremadura, 10003 Cáceres, Spain; ivanmartinmiguel97@gmail.com (I.M.-M.); diegomun@unex.es (D.M.); adescuder@alumnos.unex.es (A.E.-T.); 2Department of Sports Science, Faculty of Medicine, Health and Sports, Universidad Europea de Madrid, 28670 Madrid, Spain; info.conderipoll@gmail.com (R.C.-R.); alvaro.bustamante@universidadeuropea.es (Á.B.-S.); 3Faculty of Sport Sciences, University of Murcia, 30700 Murcia, Spain

**Keywords:** racket sports, performance, teaching, PRISMA

## Abstract

The aim of this scoping review was to examine the existing literature on padel among young players (under 18) and classify its main research areas. A systematic search in PubMed, Scopus, and Web of Science identified 16 studies on teaching methodologies, psychological characteristics, physiological demands, physical attributes, and gameplay parameters. This review provides the first comprehensive synthesis of research on youth padel. The findings suggest that a search-based teaching methodology enhances skill acquisition more effectively than traditional methods. Modifying the court dimensions (20 × 10 m to 10 × 6 m) and ball pressure optimizes learning in early training (~8–10 years). At advanced levels, training with professional players increases motivation and performance. The psychological analysis shows higher self-confidence and lower cognitive and somatic anxiety, with boys exhibiting greater somatic anxiety than girls, highlighting the need for sex-specific psychological strategies. The physiological findings establish reference values, with a higher VO2max in boys and younger players. In physical performance, boys outperform girls in terms of jump height and strength, while girls excel in agility. The gameplay analysis reveals that the rally duration increases with the skill level (7–9 s in beginners, 9–12 s in national players), the stroke frequency varies by level (from 4 at initiation level to 6–9 at regional and national levels), and there are differences in specific technical actions (forehand and backhand for initiation level, volleys for advanced level, and bandeja to finish points). From a practical standpoint, these insights can help coaches to tailor training strategies by considering a player’s age, sex, and competitive level, optimizing youth padel performance.

## 1. Introduction

Padel originated in 1969 in Mexico [1] and is one of the most widely practiced racket sports globally, with over 30 million players across more than 135 countries [2]. This sport is played in pairs (2 vs. 2) on a court measuring 20 × 10 m, divided by a central net and enclosed by mesh and glass walls [3]. These walls, which range from three to four meters in height at the back and sides, can be strategically utilized during gameplay. The dynamic nature of padel, combined with its social appeal and accessibility, makes it a compelling option in promoting health and encouraging physical activity among the population [4].

The sport has garnered significant popular interest due to its social and competitive characteristics [5], contributing to an increase in both the number of competitive players [6] and research in various domains [7]. Key research areas include game analysis, psychological factors, the physical and physiological profiles of players, and teaching methodologies [7].

### 1.1. Analysis of Game Parameters

The analysis of game parameters, particularly game actions, has emerged as the primary research area [8]. Various studies have established an average set duration of 40 min [9], with the average point durations varying by sex (13.98 s for men and 15.49 s for women) and a similar number of strokes between sexes (10–11 strokes) [10], with an increase from the first to the third set, which causes the conditional aspects of physical preparation to become more relevant [10]. Additionally, these studies have identified that the smash and forehand volley are the strokes with the most winners for men and women, respectively [11], with the volley being the most frequently used stroke for men [12] and with differences in use depending on the set, which may be due to changes in playing style due to neuromuscular and/or psychological factors [13]. These findings allow for the identification of tactical parameters with significant practical applications in training.

### 1.2. Psychological Analysis

The psychological analysis of players has become the second most researched area [7], due to the impact of competition on the players’ psycho-emotional resources [14]. In this context, there are three main effects of the emotional load. These include cognitive anxiety (negative thoughts, discomfort, and feelings of insecurity caused by the fear of negative social evaluation, failure, and loss of self-esteem [15,16]) and somatic anxiety (increased physiological activation caused by nervousness, such as an elevated heart rate, respiration, and muscle tension [17,18]), which have a negative effect on player performance. Conversely, self-confidence (the athlete’s belief in their ability to perform well in competition [19]) has a positive effect on sports performance [20]. In addition, emotional intelligence in emotional management has been demonstrated to play a crucial role in the realms of emotional management, performance under pressure, and sportsmanship, as evidenced by studies conducted on tennis and badminton [21].

### 1.3. Physiological and Physical Profiles

Similarly, the analysis of the physiological and physical profiles has provided insights into the characteristics of competition and players in other racket sports [22,23]. Internal load parameters, such as the heart rate (HR) and oxygen consumption (VO2), are used to quantify competitive efforts [24] and to identify the limits of cardiovascular function [25]. Compared to other racket sports, badminton is the sport with the highest metabolic and cardiovascular demands [26], being higher than padel in terms of the VO2max [27]. In padel specifically, both the HR [28,29] and VO2 [28,30] are influenced by the level of play and the player’s sex among senior players. Additionally, physical characteristics such as strength (maximal, vertical, and horizontal) and changes in direction, as well as the player height, differ between male and female players in amateur and professional categories [30,31,32], potentially affecting their playing styles based on these characteristics and demands.

### 1.4. Teaching Methodologies

Furthermore, the analysis of teaching methodologies in various sports disciplines aims to understand the impact of different teaching styles on the educational processes of players, particularly children and young beginners [33,34]. In padel, the use of videos for students in formative cycles has been shown to aid in study, comprehension, motivation, and the reinforcement of learning [35]. Additionally, a multi-methodological teaching approach by coaches, incorporating demonstrations and error corrections (group or individual) [36], as well as designing tasks that are adaptive, varied, innovative, and challenging for beginners [37], can enhance learning at these levels of play among senior players.

A review of the scientific literature indicates that theoretical studies have been conducted on various areas of knowledge in padel and have allowed the elaboration of different systematic reviews [8,38] that have focused on senior players (+18 years old). In recent years, the participation of younger players has increased, currently accounting for 12.5% of federative licenses in Spain among minors [6]. This sport, characterized by its dynamism and accessibility, has been integrated into school settings, generating growing interest among primary education students [39]. Furthermore, racket sports promote fundamental motor skills such as coordination and agility, positively contributing to children’s psychomotor development [40]. Similarly, when training young players, it is essential to adapt the training characteristics to avoid adverse effects on their physical adaptation, growth, and maturation [41]. Consequently, following the recommendations of the aforementioned study, the training and development approaches at these stages differ significantly, as they must be tailored to the developmental needs of younger players, prioritizing both athletic progress and the prevention of overload and other risk factors. Therefore, given the gap in the literature regarding the number of reviews that focus exclusively on senior players, and the necessary differences in the training and learning processes between young and adult players, the aim of this scoping review was to examine the existing literature on padel among young players (under 18) and to undertake the classification of the main fields of study within this research topic.

## 2. Materials and Methods

### 2.1. Study Design

This study was conducted based on the principles of a systematic review, as outlined by Cartwright-Hatton [42]. The information search was carried out using three specialized sport science databases: Web of Science, Scopus, and PubMed. The keywords used included padel AND (Young OR youth OR boys OR girls OR children OR kids OR under OR U12 OR U14 OR U16 OR U18 OR junior* OR menor* OR niños OR jóvenes). The final search was conducted on 20 December 2023, encompassing all research up to this date.

### 2.2. Inclusion and Exclusion Criteria

The selection of articles for the review was based on the following inclusion criteria: (a) sample of padel players under 18 years of age; (b) original studies; (c) variables related to methodology, teaching, and performance, especially game parameters, health, physiological, and physical aspects; (d) studies published in scientific journals (indexed in JCR or Scimago SJR or meeting at least 30 Latindex criteria); and (e) book chapters. Excluded were (a) abstracts, conference communications, doctoral dissertations, and systematic reviews; (b) variations of padel, such as individual padel, adapted padel, and wheelchair padel; and (c) samples of padel players over 18 years of age. Articles written in English and Spanish were included.

### 2.3. Identification and Selection of Studies

The process of searching the studies to be analyzed followed the PRISMA (Preferred Reporting Items for Systematic Reviews and Meta-Analyses) method. The identification of potential studies to include was based on five phases [43,44]: (i) searching for original studies in the databases, (ii) removing duplicate studies, (iii) first screening phase based on title and abstract, (iv) second screening phase based on the full text of the article, (v) search for primary sources.

### 2.4. Literature Search and Selection

Figure 1 presents the PRISMA flow diagram, illustrating the studies obtained during the search process. The keywords used in the initial search identified a total of 62 articles. After analyzing the articles, 11 were eliminated due to duplicates. Of the remaining 51, 8 were removed after the analysis of the titles and abstracts. Subsequently, of the remaining 43, 5 were removed because they were not in full text and then 7 were removed because of the quality of the studies, namely 9 because they included players over the age of 18 and 6 because they were abstracts, conference papers, and/or systematic reviews. Ultimately, 16 studies were included in the review for analysis.

### 2.5. Risk of Bias in Included Studies

The quality assessment of the articles was performed using the Cochrane Risk of Bias Tool (CROB). It consists of six domains with the following scoring: high risk, low risk, and unclear risk [45]. The methodological quality is described in Figure 2 and Figure 3. Overall, all studies were dominated by a “low risk of bias” in the proposed quality criteria. Only the criterion “blinding of participants and personnel (performance bias)” showed some concerns.

## 3. Results

The most frequently analyzed topic was gameplay parameters (n = 6), followed by physical characteristics (n = 5) and physiological, psychological, and teaching parameters (n = 3) (Table 1). The articles were published between 2011 and 2023. Of the included studies, 58.82% were published in Journal Citation Report (JCR) journals. Table 2 presents a summary of each study included in the review, indicating the sample, variables, and most relevant results.

## 4. Discussion

The aim of this scoping review was to examine the existing literature on padel among young players (under 18) and to perform the classification of the main fields of study in this research topic. Following the methodology for the searching and selection of studies, a total of 16 articles were included. They focused on the following fields of study: teaching methodologies, psychological characteristics, physiological parameters, physical characteristics, and gameplay parameters.

### 4.1. Teaching Methodologies

During the formative and initial stages, an effective teaching methodology is crucial in acquiring the necessary knowledge for learning. According to Prieto-Bermejo and Renes-López [52], any teaching style can improve the knowledge of players in the initiation stages around the age of eight. However, these authors suggest that a search-based methodology (learning in variable game situations) significantly enhances performance in baseline-to-baseline and baseline-to-net situations compared to a traditional methodology (repetition-based exercises and automatization of decontextualized actions). Additionally, the materials used in the teaching process are crucial for the student’s training. Firstly, adapting the court size from 20 × 10 m to 10 × 6 m increases the execution of wall shots, volleys, and overhead shots [49]. Secondly, while modifying the pressure of the balls does not affect temporal, physiological, or specific game actions, using lower-pressure balls for beginners around 10 years old promotes a more enjoyable, comfortable environment and facilitates game development [53]. Therefore, in the initiation stages with younger players, a search-based teaching methodology combined with modified instructional materials (court dimensions, net and racket size, and ball pressure) can lead to greater learning at these ages and levels, enhancing the educational process of the players. Additionally, Díaz-García et al. [56] indicated that exposing elite players to an external stimulus during training (e.g., a session with a professional player) increases motivation (intrinsic and extrinsic), fatigue, mental load, and physiological variables. Therefore, implementing a reward system adapted to the player’s level and age to generate appropriate demands may be an interesting approach to increase engagement and performance improvements.

### 4.2. Psychological Characteristics

Psychological characteristics can vary according to age and sex. Generally, padel players exhibit higher levels of self-confidence compared to cognitive and somatic anxiety [14,16]. Regarding sex, Rodríguez-Cayetano et al. [16] indicated that both sexes show similar values of cognitive anxiety, somatic anxiety, and self-confidence. Rodríguez-Cayetano et al. [14] confirmed the data on cognitive anxiety and self-confidence but showed differences in somatic anxiety, with boys exhibiting higher values. Therefore, these results suggest that playing padel generates greater self-confidence in players compared to somatic anxiety, which is related to physical problems, or cognitive anxiety, which is linked to negative thoughts and worries. This effect is similar in both sexes, except for somatic anxiety, which requires further investigation for validation. Furthermore, Rodríguez-Cayetano et al. [14] found a correlation between the variables, indicating that higher self-confidence is associated with lower anxiety levels, while higher somatic anxiety is associated with higher cognitive anxiety. In relation to age, younger players present better values of self-confidence, cognitive anxiety, and somatic anxiety [14]. Therefore, these data suggest that, as players age, they may experience lower self-confidence and higher levels of anxiety. Finally, compared to other racket sports like tennis, male padel players exhibit higher levels of self-confidence but also higher levels of somatic and cognitive anxiety, while female padel players exhibit better levels of self-confidence and cognitive anxiety compared to tennis players [14].

### 4.3. Physiological Parameters

In terms of physiological parameters, competition results in male players having a VO2 of 24.06 ± 6.95 mL/kg/min (43.73 ± 11.04% of VO2max obtained in the laboratory), a maximum heart rate (HRmax) of 169.72 ± 18.41 bpm (18% lower than HRmax in the laboratory), and an average heart rate (HRavg) of 73.99 ± 4.65% of HRmax in the laboratory [46]. The values reported in [47] are similar, indicating minimum heart rate (HRmin) values of 95.45 bpm, HRavg of 141.23 bpm, and HRmax of 175.24 bpm in the first set. In relation to sex and age differences, boys have higher VO2max values compared to girls, and younger players (U14 vs. U16) have higher VO2max values in both sexes [58]. Finally, padel is considered a moderate-intensity sport located at the anaerobic threshold (VT2) [46].

### 4.4. Physical Characteristics

Regarding the physical characteristics of the players, the results presented by Courel-Ibáñez and Llorca-Miralles [55] show higher values in height (172.8 vs. 158 cm), weight (70.2 vs. 54.7 kg), and body mass index (23.3 vs. 21.7) in boys compared to girls, who, in turn, have a higher body fat percentage (24.1 vs. 16.1%). The physical conditions of padel players show differences related to strength (vertical, horizontal, and throwing), direction changes, the distance covered, and flexibility according to sex. Regarding vertical strength, boys have higher jump heights than girls [55,58]. Regarding throwing strength and agility, there are discrepancies among the results. On the one hand, the data presented by Courel-Ibáñez and Llorca-Miralles [55] indicate that, in players with an average age of 14, values for the medicine ball throw (dominant arm, non-dominant arm, and overhead) are similar in both sexes. On the other hand, Sánchez-Alcaraz and Orozco-Ballesta et al. [51] show that boys have higher values in the overhead throw, while girls achieve better results in agility. Data related to horizontal strength show that, at shorter distances (10 m), girls achieve better results, whereas, at longer distances (20 m), boys perform better [51]. These authors also add a positive correlation between the linear speed and change of direction (COD) but a negative correlation between the linear speed and strength. Lastly, in the study by Pradas et al. [58], it was observed that the jump height, maximum oxygen consumption, distance covered, and flexibility variables were influenced by age in both sexes, with older players showing higher values (U16 vs. U14). The results show differences in the data provided by these studies across various variables, indicating a need for further research on these physical capacities to establish standard values for use with players, allowing coaches to assess their physical condition and adapt their training accordingly.

### 4.5. Health-Related Physical Parameters

Regarding health-related physical parameters during padel practice in young players, Delgado-García et al. [57,58] indicated that padel practice does not generate asymmetries in the lower limbs; however, it does generate asymmetries in the upper limbs, which are not affected by the maturity status [57]. Finally, injuries are a critical aspect of sports practice, with Castillo-Lozano and Casuso-Holgado [48] indicating that the most common injuries in padel for minors are low back pain (23.30%), knee sprains, plantar fasciitis, and elbow injuries (10%), as well as wrist, ankle, and shoulder sprains (6.70%). They also indicate that the body mass index, laterality, and age variables can explain 7.5–18.5% of injuries.

### 4.6. Game Indicators

Game indicators related to temporal parameters indicate that matches played by lower-level players (initiation, regional category) have an average point duration of 7–9 s [47,53,54]. On the other hand, advanced-level matches show a point duration of around 7 s in boys [46], with these values being lower than those proposed by García-Benítez et al. [50] in the national category, indicating that the point duration varies based on age and sex, ranging from 9 to 12 s in boys and 11 s in girls. They also indicate that the point duration is longer in older age categories, especially in boys. Therefore, there seem to be discrepancies regarding the point duration, but the results suggest that higher-level and older players have longer point durations.

### 4.7. Game Parameters

Regarding game parameters, a lower number of strokes per point is observed at initiation levels (four strokes per point) compared to regional (seven to nine) or national levels (six to eight) [47,50,53,54]. Additionally, they are slightly higher in older players (U18 vs. U16) [50]. The stroke frequency shows the higher use of forehand, backhand, and wall shots at initiation [54], whereas direct strokes (volleys and smashes) surpass bounce strokes (forehand, backhand) in advanced players [46]. Therefore, the data suggest more backcourt play at the initiation level and more net-to-backcourt play in higher-level players. Finally, concerning specific stroke analysis, only Sánchez-Pay et al. [59] have examined this issue focusing on under-18 padel players’ strokes. These authors indicate that the bandeja is the most used finishing move in both sexes, followed by the off-the-wall smash in girls and flat smashes in boys. Similarly, they show more winners with the flat smash in girls and topspin smash in boys and more continuity and errors with the bandeja. Regarding the stroke direction, the bandeja and topspin smash are mainly cross-court, with the flat smash down the line and the off-the-wall smash being similar, but, according to sex, girls execute it more down the line (52.5%), while boys do so more cross-court (65.4%).

### 4.8. Study Limitations and Future Studies

Research on padel has increased in recent years, covering multiple areas but focusing on professional players. The main limitation of this review has been the limited number of studies exclusively focused on younger players. To our knowledge, only those included in this review that met the inclusion criteria exist. Likewise, a problem arising from the limited research on young players is the lack of information on the different areas addressed. The results have allowed us to gain an understanding of the various research areas; however, it has not been possible to corroborate this information with more studies to draw more robust conclusions. Therefore, future studies should continue to address the analyzed parameters in younger players to enhance their development as athletes and help them to grow appropriately as players. Finally, it would be interesting for future work to include meta-analyses, and the synthesis of key findings across studies would enhance the robustness of the conclusions.

## 5. Conclusions

Training during formative ages is crucial for the development of padel players. This study provides, for the first time, a comprehensive analysis integrating psychological, physiological, physical, and technical–tactical aspects to optimize the training strategies for young athletes. The findings offer concrete guidelines for coaches and practitioners to enhance player development through evidence-based methodologies.

Specifically, our results indicate that learning in variable game situations facilitates more effective skill acquisition. Adjusting the court dimensions (reducing from 20 × 10 m to 10 × 6 m) and modifying the ball pressure can significantly benefit younger players (~8 years old) in the initial learning stages. For advanced levels, incorporating external stimuli to boost motivation enhances both performance and learning outcomes.

From a psychological perspective, our study reveals that higher self-confidence and lower cognitive and somatic anxiety correlate with better competitive performance, with younger players showing more favorable values in both sexes. Given the higher somatic anxiety levels observed in boys, training interventions should prioritize emotional regulation techniques tailored to age and gender.

Regarding physiological and physical demands, our study establishes reference values that coaches can use to design sport-specific conditioning programs. For instance, male U18 padel players exhibit VO2 levels of 24.06 ± 6.95 mL/kg/min, with heart rate metrics averaging 169.72 ± 18.41 bpm for HRmax and a HRavg of 141.23 bpm. Additionally, boys demonstrate greater heights (172.8 cm vs. 158 cm) and superior CMJ performance (23.2 cm vs. 9.9 cm) compared to girls, highlighting the necessity of individualized training strategies.

Furthermore, our study identifies key gameplay parameters that distinguish competitive levels. Lower-level players engage in rallies lasting 7–9 s, while national-level players extend their exchanges to 9–12 s. The stroke frequency increases with the competitive level, ranging from four per rally in beginners to six to nine at the regional and national levels, with U18 players executing slightly longer rallies than U16 athletes. Technically, forehands and backhands dominate the initiation stages, while volleys gain prominence at advanced levels, and the bandeja emerges as a critical stroke for point conclusion across both sexes.

By synthesizing these findings, our study provides a framework for the design of training sessions that replicate competition demands across psychological, physical, and technical–tactical dimensions. This approach would ensure that young padel players receive structured, evidence-based training to enhance their performance and long-term development.

## Figures and Tables

**Figure 1 sports-13-00075-f001:**
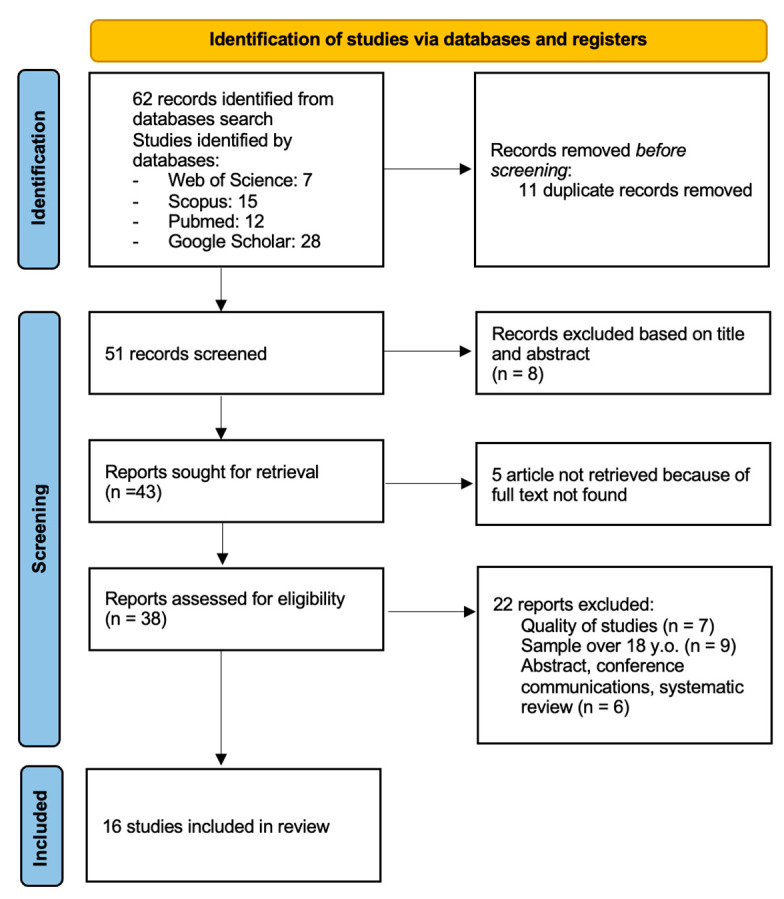
PRISMA flow diagram.

**Figure 2 sports-13-00075-f002:**
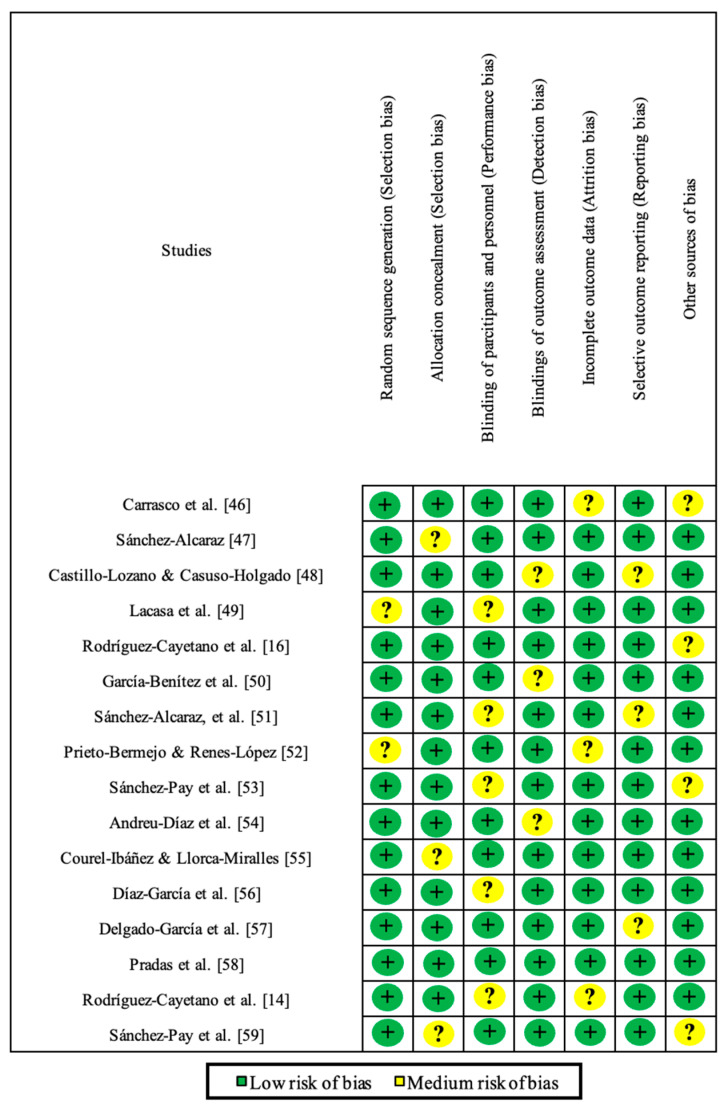
Assessment of the methodological quality of the studies using Cochrane’s Risk of Bias Tool for Randomized Trials.

**Figure 3 sports-13-00075-f003:**
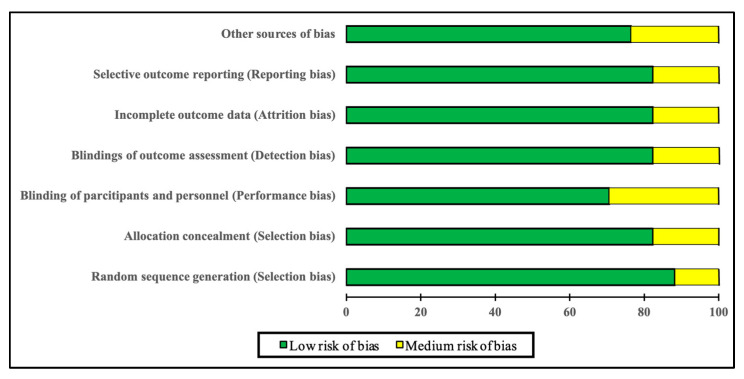
Risk-of-bias items presented as percentages across all included studies.

**Table 1 sports-13-00075-t001:** Classification of articles by study topic.

Nº	Article	Teaching	Psychology	Physiology	Physical Characteristics	Gameplay Parameters
1	Carrasco et al. [46]					
2	Sánchez-Alcaraz [47]					
3	Castillo-Lozano and Casuso-Holgado [48]					
4	Lacasa et al. [49]					
5	Rodríguez-Cayetano et al. [16]					
6	García-Benítez et al. [50]					
7	Sánchez-Alcaraz et al. [51]					
8	Prieto-Bermejo and Renes-López [52]					
9	Sánchez-Pay et al. [53]					
10	Andreu-Díaz et al. [54]					
11	Courel-Ibáñez and Llorca-Miralles [55]					
12	Díaz-García et al. [56]					
13	Delgado-García et al. [57]					
14	Pradas et al. [58]					
15	Rodríguez-Cayetano et al. [14]					
16	Sánchez-Pay et al. [59]					

Note. Gray background = the article deals with this topic of study.

**Table 2 sports-13-00075-t002:** Summary of the studies analyzed.

Nº	Authors	Sample	Variables	Results
1	Carrasco et al. [46]	12 advanced-level right-handed boys (16.57 ± 1.51 years; 1.72 ± 0.08 m; 66 ± 11.37 kg; BMI 22.24 ± 2.73 kg/m^2^).	- Physiological variables: VO2, %VO2max, Max HR, Mean HR, VT2. Measured in laboratory and competition - Temporal variables: real game time, point duration, pause between points, game–rest time ratio. - Stroke frequency variables.	VO2 in competition 24.06 ± 6.95 mL/kg/min. %VO2max 43.73 ± 11.04% in laboratory test. Max HR in laboratory 200.43 ± 15.76 bpm. Max HR in match 169.72 ± 18.41 bpm, 18% lower. Mean HR is 73.99 ± 4.65% of Max HR obtained in the laboratory. VT2 of 52.52 ± 15.50%, indicating moderate intensity in competition. Game time 163.06 s. Real game time 71.43 s. Point duration 7.24 s. Pause between points 9.11 s. Game–rest time ratio 0.97. Volley (25.57%) and smash (12.45%), most used strokes without bounce. Forehand (20.16%), smash (12.45%), and backhand (8.36%), most used strokes with bounce. Lob least used (2.95% without bounce and 1.80% with bounce).
2	Sánchez-Alcaraz [47]	16 boys (14.24 ± 1.86 years, 165.46 ± 7.45 cm, and 58.67 ± 8.93 kg). Minimum of 2 years of practice and participation in 10 tournaments per year. First set of 8 matches from the regional junior padel championship.	- Game actions: total strokes per point, from the left side and from the right side. - Temporal variables: total game time, actual playing time, rest time, mean duration of each point, mean duration of pauses between points. - HR: HRavg, HRmin, and HRmax.	Average total game time 1745.21 s (between 17 and 39 min). Average rest time 1212.98 s (20.2 min). Mean duration of each point 9.23 s. Mean pause time between points 14.12 s. Number of strokes on the right side 3.28. Number of strokes on the left side 3.45. Average total strokes per point 7–9. HRmin 95.45 bpm. HRavg 141.23 bpm. HRmax 175.24 bpm.
3	Castillo-Lozano and Casuso-Holgado [48]	30 players, 24 boys and 6 girls (17.5 ± 2.1 years, 1.75 ± 0.89 cm, 70.13 ± 11.1 kg, 22.65 ± 2.63 body mass index).	Percentage of injuries: head/neck, upper limbs, trunk, and lower limbs.	The most common injuries are low back pain (23.30%); knee sprains, plantar fasciitis, and elbow injuries (10%); wrist, ankle, and shoulder sprains (6.70%). BMI, handedness, and age variables could explain 7.5–18.5% of the injuries.
4	Lacasa et al. [49]	8 players (5 boys and 3 girls, aged 7.6 ± 0.7 years).	- Hitting: baseline, wall, net. - Situation A: nine-game match, court size 20 × 10 m. - Situation B: adapted match, court 10 × 6 m, padel length 33 cm, soft ball.	Significantly more hits against the wall (8.57 ± 2.22 vs. 3 ± 0.82) and at the net (21.25 ± 5.25 vs. 5.25 ± 1.26) in the adapted match. More defining shots in the adapted match (3.75 ± 2.06 vs. 0.50 ± 1.00) and similar baseline hits in both matches (63.25 ± 9.91 vs. 64.75 ± 15.09) were not significant. The adapted match promotes the greater use of wall shots, as well as volleys and lobs.
5	Rodríguez-Cayetano et al. [16]	221 players (100 girls, 121 boys). Age categories: U12 = 93; U14 = 73; U16 = 55.	Revised Competitive State Anxiety Inventory-2 (CSAI-2R), Spanish version. Sixteen items across three subscales: cognitive anxiety, somatic anxiety, and self-confidence. Measurement on Likert scale.	No differences found in cognitive anxiety (1.97 ± 0.758 vs. 1.95 ± 0.681), somatic anxiety (1.44 ± 0.608 vs. 1.56 ± 0.631), and self-confidence (3.18 ± 0.642 vs. 3.30 ± 0.641) between girls and boys. Differences observed between U12 (1.74 ± 0.736) and U16 (2.20 ± 0.704) in cognitive anxiety. Differences noted between U12 (3.44 ± 0.616), U14 (3.07 ± 0.608), and U16 (3.15 ± 0.650) in self-confidence. Differences observed between U16 (1.62 ± 0.623), U14 (1.49 ± 0.580), and U12 (1.45 ± 0.651) in somatic anxiety. Self-confidence values (3.24 ± 0.642) were higher than cognitive anxiety (1.96 ± 0.716) and somatic anxiety (1.51 ± 0.622) values.
6	García-Benítez et al. [50]	1670 points from 32 matches in the national youth category, 16 boys and 16 girls, U18 (15 ± 1.08 years).	- Sex. - Point duration. - Rest between points. - Strokes per point. - Lobs per point. - Effective playing time. - Work–rest ratio.	Boys have less total playtime in U16 (29%, 3454.86 s) compared to U18 (34.7%, 3404.742 s). Average of 995 and 1185 strokes per point and 169 and 259 lobs per match in U16, U18, respectively. Point duration between 8.9 and 12.0 s. Rest time of 14.3 and 15.5 s. There are between 6.1 and 8.0 strokes per point. Girls have a total playtime of 32.4% in U16 and 34.8% in U18. There are 986 points per game. Point duration is 11.3 and 11.7 s, rest time of 15.6 and 14.1 s, and between 6.9 and 7.2 strokes per point between U16 and U18, respectively.
7	Sánchez-Alcaraz et al. [51]	17 players, 8 boys (14.12 ± 1.24 years) and 9 girls (14.33 ± 1.24 years). Minimum practice of 2 days per week. Minimum participation in 10 tournaments per season.	- Speed of movement, 10 and 20 m sprint. - Change of direction and speed. Hexagon test. Upper body strength. Medicine ball throw.	Boys show better 20 m sprint times (3.87 ± 0.30 vs. 4.29 ± 0.37 s) and medicine ball throw distances (625 ± 9.70 vs. 423 ± 6.70 cm) compared to girls. Girls demonstrate better 10 m sprint times (2.84 ± 0.17 vs. 2.56 ± 0.26 s) and agility (15.39 ± 3.06 vs. 14.39 ± 2.32 s). Linear speed and change of direction variables positively correlate with each other. Strength variables correlate negatively with linear speed and change of direction.
8	Prieto-Bermejo and Renes-López [52]	45 participants at beginner level (8.62 ± 2.06 years; 21 boys and 24 girls). Experimental group (n = 22) and control group (n = 23).	- Experimental group (1): Methodology based on exploration, focused on acquiring skills in variable game situations. - Control group (2): Traditional methodology emphasizing repetition of exercises and automation of actions.	Tests for baseline (pre) and post-training for baseline-to-baseline (group 1—5.05 vs. 9.73; group 2—4.52 vs. 5.17) and baseline-to-net (group 1—2.86 vs. 4.73; group 2—3.26 vs. 3.96) demonstrate improvements in both groups, with more pronounced improvements in the group using the exploration-based teaching approach.
9	Sánchez-Pay et al. [53]	16 players (10 boys and 6 girls) at beginner level (10 ± 0.8 years; 146.0 ± 4.9 cm; 37.4 ± 7.3 kg). They trained 2 h per week, with one year of competitive experience. Four matches were played.	- Physiological variables: HRavg, %HRmax. - Psychological variables: Rating of Perceived Exertion (RPE), satisfaction questionnaire. - Game variables: set and point duration, points per set, strokes per set and per point, strokes with bounce and without bounce. - Ball type: official ball vs. low-pressure ball.	Set duration (24.03 ± 8.38 vs. 23.38 ± 5.06 min). Number of points per set (51.75 ± 20.21 vs. 48.25 ± 12.55). Number of strokes per set (223.25 ± 91.03 vs. 209 ± 51.19). Number of strokes per point (4.31 ± 2.76 vs. 4.33 ± 2.61). Strokes with bounce (3.81 ± 2.13 vs. 3.86 ± 2.18). Strokes without bounce (1.64 ± 1.20 vs. 1.49 ± 0.83). Point duration (8.07 ± 4.79 vs. 7.99 ± 4.63). Official ball vs. low-pressure ball showed no significant differences. Seventy percent of points were concluded with between 2 and 5 strokes. Thirty percent of points had more than 5 strokes. HRavg and %HRmax were higher in matches with the official ball (145 bpm, 72.5% HRmax) compared to low-pressure balls (140 bpm, 69.9% HRmax). There were no differences in subjective perception of effort. Thirty percent of game time with low-pressure balls and 20% with official balls were in low- to very low-intensity zones. Moreover, 12.82 min with normal balls and 10.94 min with low-pressure balls were spent in moderate-intensity zones (65–76% HRmax). Greater enjoyment, comfort, and ease of play were reported with low-pressure balls.
10	Andreu-Díaz et al. [54]	Eight male beginner players (13.5 ± 1.4 years; 1.51 ± 0.21 m; 46.25 ± 4.51 kg) participated in two matches, totaling 723 strokes (176 points). They had a minimum of one year of training experience, with two hours of training per week.	- Temporal variables: average point duration, average rest duration between points. - Game action variables: number of strokes, number of points, type of stroke, stroke direction, stroke effectiveness, court side.	Strokes per point 4.08 ± 2.73. Strokes per game 25.07 ± 10.69. Strokes per set 242.33 ± 76.14. Strokes per match 363.5 ± 13.5. Points per game 6.07 ± 2.36. Points per set 58.66 ± 10.53. Points per match 88 ± 15. Average duration 7.66 s, constituting 33% of total game time. Rest duration 15.47 s, comprising 67% of total game time. Primary strokes included first serve (18.4%), forehand return (18.2%), forehand (17.8%), and forehand wall (8.9%). The winning pair utilized more backhand volleys (2.3 vs. 0.8%) and forehand wall (10.1 vs. 7.7%) and less forehand volleys (5.1 vs. 6.6%) and forehand (15.8 vs. 19.5%) compared to the losing pair. The side of play and stroke direction did not determine the outcome. The winning and losing pairs executed 30.4% and 22.6% more cross-court strokes. The left-side player executed 11% more down-the-line strokes than the right-side player. The right-side player had 10–16.4% more involvement in both pairs. Errors accounted for 15–19%, and winners for 6.5–7.3%. The side of play did not determine stroke effectiveness.
11	Courel-Ibáñez and Llorca-Miralles [55]	34 players (19 boys and 15 girls): 14.6 ± 1.5 years old; 63.4 ± 14.5 kg; height 166.6 ± 9.8 cm; 6.2 ± 2.5 years of experience. Regional category of the Andalusian Padel Federation.	- Sex - Anthropometry - Body composition - Change in direction and agility - Jump and strength test	Boys had an average height of 172.8 cm, weight of 70.2 kg, 16.1% body fat, and BMI of 23.3. Girls had an average height of 158 cm, weight of 54.7 kg, 24.1% body fat, and BMI of 21.7. Boys’ jump height (CMJ = 23.2 cm; Abalakov = 27 cm) was higher than girls’ (CMJ = 9.9 cm; Abalakov = 11.7 cm). Strength values (medicine ball throw dominant hand = 4.7–5 m; non-dominant hand = 4.8 m; overhead throw = 5.2–6 m) and change in direction (padel agility test = 18.2–19.4 s; 3 × 10 m sprint = 8.3–8.8 s) showed no differences between sexes.
12	Díaz-García et al. [56]	36 elite players (22 boys, 17.40 ± 2.16 years old; 14 girls, 17.90 ± 3.21 years old) participated in four matches, two of which had rewards.	- Situational Motivation Scale - Heart rate variability - Quantification of mental load - Visual analog scale - Psychomotor vigilance task	Intrinsic motivation (6.31 ± 1.98 vs. 5.17 ± 1.47), external motivation (4.45 ± 1.29 vs. 3.86 ± 1.17), mental fatigue (7.08 ± 2.29 vs. 5.67 ± 1.28), and mental load (8.67 ± 2.27 vs. 6.45 ± 1.49) were higher in matches with rewards compared to matches without rewards. Reaction time (0.49 ± 0.02 vs. 0.43 ± 0.02), HRavg (146.19 ± 16.43 vs. 137.67 ± 14.89), SDNN (31.21 ± 6.91 vs. 38.76 ± 8.82), NN50 (6.52 ± 1.23 vs. 10.22 ± 1.78), and %rMSSD (19.48 ± 5.91 vs. 22.65 ± 6.98) were higher in matches with rewards compared to matches without rewards. Amotivation (2.21 ± 0.94 vs. 2.67 ± 0.98) was lower in matches with rewards compared to matches without rewards.
13	Delgado-García et al. [57]	96 players (53 boys and 43 girls) and a control group of 76 alpine skiers (43 boys and 33 girls).	- Lower limb asymmetry variable. - State of maturity.	There are no lower limb asymmetries observed in padel players, with a measurement of 1.1 ± 0.8%. However, there are upper limb asymmetries observed at 7.2 ± 5%, which is higher compared to the control group, who have values of 1.4 ± 3.2%. There are also no significant differences observed in terms of maturity status among children with a negative or positive maturity status.
14	Pradas et al. [58]	60 young padel players divided into U14 (15 boys and 15 girls) and U16 (15 boys and 15 girls).	- Physiological variables: VO2max. - Physical variables: SJ, CMJ, flexibility, distance covered, average speed, asymmetries.	VO2max in U14 (47.21 ± 4.49 vs. 41.29 ± 4.35 mL/kg/min) and U16 (45.70 ± 2.34 vs. 39.85 ± 2.73 mL/kg/min) is higher in boys compared to girls. VO2max in absolute terms (U14: 2.57 ± 0.41 vs. U16: 2.92 ± 0.34 l/min) also shows higher values in boys compared to girls. Boys demonstrate higher values in jump tests for power in SJ (U14: 1765 ± 414 vs. U16: 2388 ± 397 W) and CMJ (U14: 2002 ± 398 vs. U16: 2555 ± 382 W) than girls for SJ (U14: 1565 ± 277 vs. U16: 1724 ± 246 W) and CMJ (U14: 1741 ± 289 vs. U16: 1894 ± 263 W) and for jump height in SJ (U14: 22.9 ± 5.12 vs. U16: 25.53 ± 3.85 cm) and CMJ (U14: 25.68 ± 4.75 vs. U16: 27,72 ± 3,67 cm) than girls for SJ (U14: 21.22 ± 2.44 vs. U16: 21.16 ± 2.41 cm) and CMJ (U14: 23.70 ± 2.52 vs. U16: 23.27 ± 2.85 cm). Distance covered in shuttle run tests (U14: 1129 ± 322 vs. U16: 1203 ± 160 m) and average speed (U14: 11.37 ± 0.8 vs. U16: 11.55 ± 0.40 km/h) are slightly higher in U16 compared to U14 for both sexes. Flexibility measurements indicate higher values in U16 compared to U14 for both boys (U14: 18.91 ± 6.38 vs. U16: 25.46 ± 8.94 cm) and girls (U14: 29.31 ± 6.24 vs. U16: 32.42 ± 7.85 cm). Upper limb asymmetries are observed but are not significant between U14 (boys: 11.31 ± 5.09% vs. girls: 5.82 ± 1.65%) and U16 (boys: 10.86 ± 2.85% vs. girls: 6.75 ± 1.42%). There are no asymmetries observed in the lower limb during lateral movement, acceleration, and reaction time tests. Strength values in the dominant and non-dominant hand show similarities across ages but differ between sexes, with higher values observed in boys.
15	Rodríguez-Cayetano et al. [14]	423 players (15.40 ± 3.43 years old), including 291 padel (191 boys; 100 girls; 93 U14; 93 U16; 105 senior category) and 132 tennis players (85 boys and 47 girls; 31 U14; 34 U16; 67 senior category).	Revised Competitive State Anxiety Inventory-2 (CSAI-2R), Spanish version. Three subscales: cognitive anxiety, somatic anxiety, and self-confidence.	Higher values of self-confidence (3.25 ± 0.548) compared to cognitive anxiety (2.01 ± 0.679) and somatic anxiety (1.60 ± 0.557) are observed. Boys exhibit self-confidence (3.27 ± 0.54), somatic anxiety (1.69 ± 0.56), and cognitive anxiety (2.04 ± 0.68). Girls demonstrate self-confidence (3.22 ± 0.56), somatic anxiety (1.44 ± 0.52), and cognitive anxiety (1.96 ± 0.67). Somatic anxiety values are significantly higher in boys compared to girls. Boys show higher values in all three variables compared to tennis players. Girls exhibit lower levels of cognitive anxiety and higher levels of self-confidence than in tennis. Self-confidence is higher in U14 (3.44 ± 0.54) compared to U16 (3.27 ± 0.49) and seniors (3.07 ± 0.55). Somatic anxiety is higher in seniors (1.82 ± 0.54) than in U14 (1.44 ± 0.55) and U16 (1.52 ± 0.51). Cognitive anxiety is higher in seniors (2.26 ± 0.69) compared to U14 (1.79 ± 0.70) and U16 (1.96 ± 0.55). There is a positive correlation between cognitive anxiety and somatic anxiety (0.487) and a negative correlation between both (−0.312; −0.254) and self-confidence.
16	Sánchez-Pay et al. [59]	175 smashes from six junior national category finals (three male and three female). 12 boys and 12 girls (aged 16–18 years). Top 10 national junior category.	- Sex. - Type of smash: flat smash, topspin smash, bandeja, off-the-wall smash. - Direction of shot. - Effectiveness of smash.	Bandeja (RTC = 13) and topspin smashes (RTC = 5.7) are predominantly cross-court. Flat smashes (RTC = 19.1) are mainly down the line. Off-the-wall smashes are similar in both directions, but girls tend to execute them more down the line (52.5%), while boys tend to do so more cross-court (65.4%). Flat smashes (RTC = 9.4) and topspin smashes (RTC = 7.1) mainly result in winners. Off-the-wall smashes primarily lead to errors (RTC = 3.1), while bandejas lead to more continuity (RTC = 11.6). Regarding sex, bandejas (boys (88.5%) and girls (79.3%)) and off-the-wall smashes (boys (80.8%) and girls (61.3%)) promote continuity in the game, as do flat smashes in boys (62.8%) and topspin smashes in girls (54.5%). Flat smashes are often winners for girls (53.3%), while topspin smashes are winners for boys (50.7%) and less often result in continuity (46.3%). Bandejas are the most used shots among girls (44.1%) and boys (43.8%). Girls execute more off-the-wall smashes (13.8%) and flat smashes (36.5%) compared to boys. Boys perform more topspin smashes (22.6%) than girls.

m: meters; cm: centimeters; kg: kilograms; kg/m^2^: kilograms per square meter; BMI: body mass index; s: seconds; min: minutes; VO2: oxygen uptake; %VO2max: percentage of maximal oxygen consumption; HR: heart rate; HRmax: maximum heart rate; %HRmax: percentage of maximum heart rate; HRavg: average heart rate; HRmin: minimum heart rate; bpm: beats per minute; VT2: anaerobic ventilatory threshold; %: percentage; mL/kg/min: milliliters per kilogram per minute; W: watts; km/h: kilometers per hour; SDNN: standard deviation of normal-to-normal RR intervals; NN50: number of pairs of successive RR intervals differing by more than 50 ms, divided by the total number of RR intervals; %rMSSD: square root of the mean of the sum of the squares of differences between adjacent RR intervals; U12: under 12 years; U14: under 14 years; U16: under 16 years; U18: under 18 years; RPE: rating of perceived exertion; CMJ: countermovement jump; SJ: squat jump; RTC: residual typified corrected.

## Data Availability

The original contributions presented in the study are included in the article; further inquiries can be directed to the corresponding author.

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
