# Peer review of "Topics of Study in Under-18 Padel Categories: A Scoping Review"

_sports, 2025, doi:10.3390/sports13030075_

Round 1
Reviewer 1 Report
Comments and Suggestions for Authors
Topics of study in under-18 padel categories: A systematic review
Interesting, actual and with practical benefits work. The manuscript in my opinion is valuable and practical significant. The manuscript structure is excellent and it is obvious that the Authors have done an enough work. Personally, I learned important information. I propose some notes to improve the manuscript quality.
1. Introduction
Тhis paragraph is well-formed in terms of meaning, but it is not in terms of depth. For me, it is necessary to analyze and include more literary sources for kinematics game parameters and their relations (correlations) with other characteristics.
2. Materials and Methods
The number of papers included in the study is enough. I want to congratulate the authors for the clear and excellent structuring of this paragraph.
3. Results
In my opinion this section is excellent presented in accordance to the obtained data structure.
4. Discussion
This work has its limitations which are described here.
The Discussion is laid out consistently and clearly.
5. Conclusion.
A well-formed conclusion, but with one flaw in my opinion. Authors do well to include some self-promotion - what did they do for the first time? And specifically list the concrette benefits of their research from the viewpoint of sport practice. The current conclusions are a little bit а broad statements.
6. The Abstract can be improved.
Improvement. In the Abstract must be underline clearly the new results obtained from the authors which differ from those obtained till now. It must be underline the main authors contributions. The current concrete conclusions from the viewpoint of sport practice can be added in the Abstract.
Author Response
First, we would like to than the reviewer for the time and effort spent on this article, contributing to the improvement of the work. We are also very pleased to receive a first positive evaluation for its publication, if the changes made seem appropriate. We would also like to thank the reviewers for their suggestions for improvement. In the following section you will find specific answers to the suggestions followed by each reviewer's comment. Likewise, you will find the changes made in RED colour in the manuscript.
Reviewer 1
Reply to specific comments:
Introduction
- Reviewer: Тhis paragraph is well-formed in terms of meaning, but it is not in terms of depth. For me, it is necessary to analyze and include more literary sources for kinematics game parameters and their relations (correlations) with other characteristics.
Authors: Thanks for your feedback. We have included more information in the introduction section.
Materials and Methods
- Reviewer: The number of papers included in the study is enough. I want to congratulate the authors for the clear and excellent structuring of this paragraph
Authors: We sincerely appreciate your valuable feedback.
Results
- Reviewer: In my opinion this section is excellent presented in accordance to the obtained data structure.
Authors: We would like to express my gratitude once again for your constructive comments.
Discussion
- Reviewer: This work has its limitations which are described here.
The Discussion is laid out consistently and clearly.
Authors: Thank you for your comments.
Conclusion
- Reviewer: A well-formed conclusion, but with one flaw in my opinion. Authors do well to include some self-promotion - what did they do for the first time? And specifically list the concrette benefits of their research from the viewpoint of sport practice. The current conclusions are a little bit а broad statements
Authors: The section on conclusions has been modified, we hope that we have met the requirements indicated.
Abstract
- Reviewer: In the Abstract must be underline clearly the new results obtained from the authors which differ from those obtained till now. It must be underline the main authors contributions. The current concrete conclusions from the viewpoint of sport practice can be added in the Abstract.
Authors: The abstract has been modified considering your valuable comments.
Reviewer 2 Report
Comments and Suggestions for Authors
Line 43-45
Additionally, a systematic review of the TESEO database was conducted to compile doctoral theses on padel (11).
Line 104-106
(d) studies published in scientific journals (indexed in JCR, 104 Scimago SJR, or meeting at least 30 Latindex criteria); (e) book chapters. Excluded were 105 (a) abstracts, conference communications, and systematic reviews; (b)
Comment 1
Doctoral dissertations are not mentioned in the criteria for inclusion or exclusion of studies.
Line 70
In padel, both HR (27,28) and VO2 (27,29) are influenced by the level of play and sex among senior players.
Line 80-82
Additionally, a multi-methodological teaching approach by coaches, incorporating demonstrations and error corrections (group or individual) (35), as well as designing tasks that are adaptive, varied, innovative, and challenging for beginners (36), can enhance learning at these levels of play among senior players.
Line 84-88
A review of the scientific literature indicates that theoretical studies have been conducted on various areas of knowledge in padel. However, there is still a need to better analyze existing knowledge in youth categories and minors (under 18). Despite its importance, given that 12.5% of federative licenses in Spain belong to minors (2), there is a lack of systematic revisions in this field.
Comment 2
Most of the introduction refers to senior sports and the topic is under 18.
Is the only reason you are writing this study that there is a lack of systematic reviews in the under 18 age group?
Why it is important to analyze what is written in the scientific literature in this age group.
Children as young as 6 years old are also under 18. Did the study set a lower age limit? Paper applies to all age categories except seniors, why?
The introduction should better explain why such a study is needed and the focus of the introduction should relate to the age being studied.
Line 114-115
(v) forward searching (references cited in the included studies), and (vi) backward searching (citations of the included studies).
Comment 3
Nowhere is it mentioned which studies were included in this review by searching forwards and backwards.
Comment 4
Is the study registered in the register of systematic reviews?
Comment 5
Systematic reviews should cover a narrow area and go more in depth. This study covers all areas of research on this sport. In my opinion, this study is more scoping than a systematic review.
Author Response
First, we would like to than the reviewer for the time and effort spent on this article, contributing to the improvement of the work. We are also very pleased to receive a first positive evaluation for its publication, if the changes made seem appropriate. We would also like to thank the reviewers for their suggestions for improvement. In the following section you will find specific answers to the suggestions followed by each reviewer's comment. Likewise, you will find the changes made in RED colour in the manuscript.
Reviewer 2
Reply to specific comments:
- Reviewer: Doctoral dissertations are not mentioned in the criteria for inclusion or exclusion of studies.
Authors: Done.
- Reviewer: Most of the introduction refers to senior sports and the topic is under 18.
Is the only reason you are writing this study that there is a lack of systematic reviews in the under 18 age group?
Why it is important to analyze what is written in the scientific literature in this age group.
Children as young as 6 years old are also under 18. Did the study set a lower age limit? Paper applies to all age categories except seniors, why?
The introduction should better explain why such a study is needed and the focus of the introduction should relate to the age being studied.
Authors: Thanks for your feedback. We have included more information mainly in the last paragraph of the introductions. Regarding age limitation, we did not include children under 6 years old due we did not find any study with these ages.
- Reviewer: Nowhere is it mentioned which studies were included in this review by searching forwards and backwards
Authors: We agree with this comment, and we have included this information.
- Reviewer: Is the study registered in the register of systematic reviews?
Authors: We have completed the PROSPERO template to register the systematic review.
- Reviewer: Systematic reviews should cover a narrow area and go more in depth. This study covers all areas of research on this sport. In my opinion, this study is more scoping than a systematic review.
Authors: We agree, so we have modified the different sections (tittle, objectives, etc.) as a scoping review.
Reviewer 3 Report
Comments and Suggestions for Authors
Comments
The article systematically reviews a relatively new and under-researched sport, padel, providing a foundational exploration of teaching methodologies, psychological factors, physiological demands, physical characteristics, and game parameters for youth players. Its focus on a specific population (U18) and the categorization of results by age and sex offer potentially useful insights for coaches and sports scientists. The summary of training adaptations (e.g., court size modifications, use of low-pressure balls) and gameplay parameters is practical for applied use in training settings.
Key Issues and Suggestions for Improvement
Lack of Justification for Focusing on U18 Players
The article does not adequately justify why the systematic review focuses exclusively on youth players (U18). While tailoring training to youth athletes is important, the authors miss an opportunity to explain why research on this group is prioritized over an all-ages perspective, especially given the infancy of the sport and its limited existing literature.
The authors should either provide a clear rationale for focusing solely on U18 players or consider expanding the systematic review to include a broader age range. If age-related differences emerge, these could then be discussed in-depth to highlight developmental considerations.
Contextualization of Padel as a Sport
Padel is not a well-known sport globally, and many readers (myself included) may be unfamiliar with its rules, structure, and prevalence. The article assumes familiarity with the sport but provides no context for its development, popularity, or organizational structure (e.g., the number of national governing bodies).
The introduction should include a brief overview of the sport, its origins, and its current global reach, potentially citing the number of countries with official padel governing bodies. This would situate the review and make it accessible to a wider audience. Additionally, contrasting padel with similar racquet sports (e.g., tennis, squash, badminton) would add value and highlight its unique features and challenges.
Originality and Broader Comparisons
While the authors’ focus on padel shows originality, their narrow lens misses the opportunity to compare findings with more established racquet sports where research is more extensive. For instance, how do the psychological or physiological demands of padel compare with tennis or squash?
By integrating comparisons to similar sports, the authors could offer a more nuanced understanding of padel’s distinct demands while situating the findings in a broader sports science context.
Discussion
The broad aim of the review results in a lack of depth in the discussion. While the article provides an overview of various parameters, it struggles to tie these findings into a coherent narrative, making the discussion feel fragmented.
A more focused research question or subdividing the review into distinct themes (e.g., psychological, physiological, gameplay) with dedicated discussions for each could improve clarity. Additionally, including a meta-analysis or synthesizing key findings across studies would enhance the robustness of the conclusions.
My suggetstion therefore is to redo the Systematic Review
This article provides a helpful preliminary exploration of youth padel but requires substantial revisions to better justify its focus, contextualize the sport, and provide a clearer and more comprehensive discussion of its findings. Broadening the scope of the review or situating the research within a comparative framework could significantly enhance its academic and practical relevance. By addressing these issues, the authors would contribute more meaningfully to both the academic understanding of padel and its practical application in coaching and player development.
Author Response
First, we would like to than the reviewer for the time and effort spent on this article, contributing to the improvement of the work. We are also very pleased to receive a first positive evaluation for its publication, if the changes made seem appropriate. We would also like to thank the reviewers for their suggestions for improvement. In the following section you will find specific answers to the suggestions followed by each reviewer's comment. Likewise, you will find the changes made in RED colour in the manuscript.
Reviewer 3
Reply to specific comments:
- Reviewer: The article does not adequately justify why the systematic review focuses exclusively on youth players (U18). While tailoring training to youth athletes is important, the authors miss an opportunity to explain why research on this group is prioritized over an all-ages perspective, especially given the infancy of the sport and its limited existing literature.
Authors: Thanks for your comments. We have modified the introduction to justify why the systematic review focuses exclusively on youth players (U18).
- Reviewer: The authors should either provide a clear rationale for focusing solely on U18 players or consider expanding the systematic review to include a broader age range. If age-related differences emerge, these could then be discussed in-depth to highlight developmental considerations.
Authors: We have included this information in the last paragraph of the introduction.
- Reviewer: Padel is not a well-known sport globally, and many readers (myself included) may be unfamiliar with its rules, structure, and prevalence. The article assumes familiarity with the sport but provides no context for its development, popularity, or organizational structure (e.g., the number of national governing bodies).
The introduction should include a brief overview of the sport, its origins, and its current global reach, potentially citing the number of countries with official padel governing bodies. This would situate the review and make it accessible to a wider audience. Additionally, contrasting padel with similar racquet sports (e.g., tennis, squash, badminton) would add value and highlight its unique features and challenges.
Authors: We have included this information in at the beginning of the introduction.
- Reviewer: While the authors’ focus on padel shows originality, their narrow lens misses the opportunity to compare findings with more established racquet sports where research is more extensive. For instance, how do the psychological or physiological demands of padel compare with tennis or squash?
By integrating comparisons to similar sports, the authors could offer a more nuanced understanding of padel’s distinct demands while situating the findings in a broader sports science context
Authors: We understand that we could compare with other racket sports. However, due this is a review of padel studies, we wanted to highlight the main results of the studies we have included, comparing the different ages and variables in the discussion section.
- Reviewer: The broad aim of the review results in a lack of depth in the discussion. While the article provides an overview of various parameters, it struggles to tie these findings into a coherent narrative, making the discussion feel fragmented.
A more focused research question or subdividing the review into distinct themes (e.g., psychological, physiological, gameplay) with dedicated discussions for each could improve clarity. Additionally, including a meta-analysis or synthesizing key findings across studies would enhance the robustness of the conclusions.
My suggetstion therefore is to redo the Systematic Review
Authors: We agree with this comment, so we have included it as a limitation of our study.
- Reviewer: This article provides a helpful preliminary exploration of youth padel but requires substantial revisions to better justify its focus, contextualize the sport, and provide a clearer and more comprehensive discussion of its findings. Broadening the scope of the review or situating the research within a comparative framework could significantly enhance its academic and practical relevance. By addressing these issues, the authors would contribute more meaningfully to both the academic understanding of padel and its practical application in coaching and player development.
Authors: We really appreciate your valuable comments. We have tried to improve the manuscript following the reviewers’ recommendations, including more a contextualization of the sport and providing more information about the topic of the study.
Round 2
Reviewer 2 Report
Comments and Suggestions for Authors
The paper has been significantly improved and can be published.
Reviewer 3 Report
Comments and Suggestions for Authors
The authors have done a very good job revising the study and it is now ready for publication.